# FairSeg: A Large-Scale Medical Image Segmentation Dataset for Fairness Learning Using Segment Anything Model with Fair Error-Bound Scaling

**Yu Tian**[1][*] **Min Shi**[1][*] **Yan Luo**[1][*] **Ava Kouhana**[1] **Tobias Elze**[1] **Mengyu Wang**[1]
[1]Harvard Ophthalmology AI Lab, Harvard University
{ytian11,mshi6,yluo16,akouhana,tobias_elze,mengyu_wang}@meei.harvard.edu

## Abstract

Fairness in artificial intelligence models has gained significantly more attention in recent years, especially in the area of medicine, as fairness in medical models is critical to people's well-being and lives. High-quality medical fairness datasets are needed to promote fairness learning research. Existing medical fairness datasets are all for classification tasks, and no fairness datasets are available for medical segmentation, while medical segmentation is an equally important clinical task as classifications, which can provide detailed spatial information on organ abnormalities ready to be assessed by clinicians. In this paper, we propose the first fairness dataset for medical segmentation named Harvard-FairSeg with 10,000 subject samples. In addition, we propose a fair error-bound scaling approach to reweight the loss function with the upper error-bound in each identity group, using the segment anything model (SAM). We anticipate that the segmentation performance equity can be improved by explicitly tackling the hard cases with high training errors in each identity group. To facilitate fair comparisons, we utilize a novel equity-scaled segmentation performance metric to compare segmentation metrics in the context of fairness, such as the equity-scaled Dice coefficient. Through comprehensive experiments, we demonstrate that our fair error-bound scaling approach either has superior or comparable fairness performance to the state-of-the-art fairness learning models. The dataset and code are publicly accessible via https://ophai.hms.harvard.edu/datasets/harvard-fairseg10k.

## 1 Introduction

As the use of artificial intelligence grows in medical image diagnosis, it's vital to ensure the fairness of these deep learning models and delve into hidden biases that could arise in complex real-world scenarios. Regrettably, machine learning models can inadvertently incorporate sensitive attributes (like race and gender) associated with medical images, which could influence the model's ability to differentiate abnormalities. This challenge has spurred significant efforts to investigate biases, champion fairness, and launch new datasets in the fields of machine learning and computer vision.

To date, only a few public fairness datasets have been proposed for studying the fairness classification Dressel & Farid (2018); Asuncion & Newman (2007); Wightman (1998); Miao (2010); Kuzilek et al. (2017); Ruggles et al. (2015); Zhang et al. (2017); Zong et al. (2022); Irvin et al. (2019); Johnson et al. (2019); Kovalyk et al. (2022). Predominantly, most of those datasets Dressel & Farid (2018); Asuncion & Newman (2007); Wightman (1998); Miao (2010); Kuzilek et al. (2017); Ruggles et al. (2015); Zhang et al. (2017) consist of tabular data, making them ill-suited for developing fair computer vision models that demand imaging data. This gap is concerning, especially considering the rise of impactful

---

[*]Contributed equally as co-first authors

deep-learning models dependent on such data. In the field of medical imaging, only a handful of datasets Irvin et al. (2019); Johnson et al. (2019); Kovalyk et al. (2022) have been used for fairness learning. However, most of these datasets were not explicitly crafted with fairness modeling. They often include only a limited range of sensitive attributes, like age, gender, and race, thus constricting the scope of examining fairness among varied demographic groups. Additionally, they also lack a thorough benchmarking framework. More importantly, while those previous datasets and approaches offer solutions for medical classification, they overlook the arguably more critical field of medical segmentation.

Medical segmentation Liu et al. (2022); Tian et al. (2023a); Chen et al. (2021); Tian et al. (2023b); Zhou et al. (2018); Wang et al. (2022a) provides detailed spatial information on specific anatomical structures, essential for personalized treatments, monitoring disease progression, and enhancing diagnostic accuracy. In contrast to classification's broad categorization, segmentation's precise delineation offers in-depth information about organ abnormalities, facilitating better patient care from diagnosis to therapeutic interventions. Despite its significance, there's a glaring absence of public datasets for studying fairness in medical segmentation across diverse sensitive attributes. Given the importance of ensuring fairness in medical segmentation and the special characteristics of medical data, we argue that a large-scale medical segmentation dataset that is designed to study fairness and provide a playground for developing algorithmic debiasing approaches is crucial. However, creating such a new benchmark for fairness learning presents multiple challenges. First, there's a scarcity of large-scale, high-quality medical data paired with manual pixel-wise annotations, both of which are labor-intensive and time-consuming to collect and annotate. Second, existing methods Park et al. (2022); Wang et al. (2022b); Beutel et al. (2017) are primarily designed for medical classification, and the performance remains questionable when adapting to segmentation tasks. It is also uncertain whether the unfairness that exists in the segmentation tasks can be effectively alleviated by algorithms. Lastly, a universal metric for evaluating the fairness of medical segmentation models remains elusive. Furthermore, adapting existing fairness metrics designed for classification to segmentation tasks can also be challenging.

In order to address these challenges, we propose the **first** large-scale fairness dataset for medical segmentation, named Harvard-FairSeg, which is designed for fairness optic disc and cup segmentation from SLO fundus images for diagnosing glaucoma, as shown in Figure 1. Glaucoma, a prominent cause of irreversible global blindness, has a prevalence of 3.54% in the 40-80 age bracket, affecting roughly 80 million individuals Tham et al. (2014); Luo et al. (2023b); Shi et al. (2023a); Luo et al. (2023a); Shi et al. (2023c). Despite its significance, early glaucoma often remains asymptomatic, emphasizing the need for timely professional tests. Accurate segmentation of the optic disc and cup is crucial for early glaucoma diagnosis by healthcare professionals. It's important to note that Black individuals face a doubled risk of developing glaucoma compared to other groups, yet the segmentation accuracy is often lowest for this demographic. This motivates us to curate a dataset for studying segmentation fairness issues before the practical use of any segmentation models in the real world. Particularly, the highlights of our proposed Harvard-FairSeg dataset are as follows: (1) The first fairness learning dataset for medical segmentation. The dataset provides optic disc and cup segmentation with SLO fundus imaging data; (2) The dataset is equipped with six sensitive attributes collected from real-world clinical scenarios for the study of fairness learning problem; (3) We evaluate multiple SOTA fairness learning algorithms on our proposed new dataset with various segmentation performance metrics including Dice coefficient and intersection over union (IoU).

In addition to our valuable dataset, we develop a fair error-bound scaling (FEBS) approach as an add-on contribution to demonstrate that the fairness challenges in medical segmentation can indeed be tackled. The core idea of our FEBS approach is to rescale the loss function with the upper training error-bound of each identity group. The rationale is that the hard cases in each identity group may be the driving factors for the underlying performance disparities, and the hard cases in each identity group may be due to pathophysiological and anatomical differences between identity groups. For instance, Asians have more angle-closure glaucoma compared with Whites, and Blacks have a larger cup-to-disc ratio compared with other races. Explicitly tackling the hard cases by using upper error-bound in each identity group may help reduce model performance inequity. We

Table 1: Public Fairness Datasets Commonly Used in Medical Imaging.

| Dataset | Modality | Sensitive Attribute | No. of Images | Segmentation |
|---------|----------|---------------------|---------------|--------------|
| CheXpert | Chest X-ray (2D) | Age, Sex, Race | 222,793 | × |
| MIMIC-CXR | Chest X-ray (2D) | Age, Sex, Race | 370,955 | × |
| PAPILA | Fundus Image (2D) | Age, Sex | 420 | × |
| HAM10000 | Skin Dermatology (2D) | Age, Sex | 9,948 | × |
| Fitzpatrick17k | Skin Dermatology (2D) | Skin type | 16,012 | × |
| OL3I | Heart CT (2D) | Age, Sex | 8,139 | × |
| COVID-CT-MD | Lung CT (3D) | Age, Sex | 308 | × |
| OCT | SD-OCT (3D) | Age | 384 | × |
| ADNI 1.5T | Brain MRI (3D) | Age, Sex | 550 | × |
| ADNI 3T | Brain MRI (3D) | Age, Sex | 110 | × |
| **Harvard-FairSeg** | SLO Fundus Image (2D) | Age, Gender, Race, Ethnicity, Preferred Language, and Marital Status | 10,000 | ✓ |

subsequently integrate our proposed FEBS (Fair Error-Bound Scaling) approach with the recent segmentation foundation model, the Segment Anything Model (SAM) Kirillov et al. (2023), to explore whether FEBS enhances segmentation fairness across various sensitive attributes.

To facilitate the comparison between different fairness learning models, we propose equity-scaled performance metrics. More specifically, for instance, the ES-Dice is calculated as the overall Dice coefficients divided by the summation of the relative disparity between the overall Dice coefficients and the group Dice coefficients. This equity-scaled segmentation performance metric provides a more straightforward evaluation and is easier to interpret by clinicians than existing fairness metrics such as demographic parity difference (DPD) and difference in equalized odds (DEOdds) .

Our core contributions are summarized as follows:

- **Major:** We introduce the first fairness dataset for medical segmentation and benchmarked it with the state-of-the-art (SOTA) fairness learning approaches.
- **Minor 1:** We develop a novel fair error-bound scaling approach to improve segmentation performance equity.
- **Minor 2:** We design a new equity-scaled segmentation performance metric to facilitate fair comparisons between different fairness learning models for medical segmentation.

## 2 RELATED WORK

**Medical Fairness Datasets.** Healthcare disparities in disadvantaged minority groups have been a significant concern due to heightened disease susceptibility and underdiagnosis. Although deep learning provides solutions for addressing this by automating disease detection, it is essential to address the potential for performance biases. Existing medical fairness datasets such as CheXpert Irvin et al. (2019), MIMIC-CXR Johnson et al. (2019), and Fitzpatrick17k Groh et al. (2021) focus on image classification and often neglect the vital domain of medical segmentation. Additionally, their limited set of sensitive attributes like age, sex, and race reduces versatility in fairness system development. In this work, we introduce the first segmentation-focused medical fairness dataset, encompassing attributes like age, gender, race, and ethnicity in segmentation tasks.

**Fairness Learning.** Biases in computer vision datasets arise from data inequalities, leading to skewed predictions. Strategies to mitigate these biases are categorized into pre-processing, in-processing, and post-processing. Pre-processing methods, like those in Quadrianto et al. (2019); Ramaswamy et al. (2021); Zhang & Sang (2020); Park et al. (2022), de-bias training data, but may compromise computational efficiency. In-processing methods integrate fairness during model training but might sacrifice accuracy due to loss function manipulations Beutel et al. (2017); Roh et al. (2020); Sarhan et al. (2020); Zafar et al. (2017); Zhang et al. (2018); Shi et al. (2023b). Post-processing methods offer corrective measures but have limitations during testing Wang et al. (2022b); Kim et al. (2019); Lohia et al. (2019). Our work proposes a novel fair error-bound scaling approach, hypothesizing that addressing hard cases within identity groups could enhance model fairness performance.

**Fairness Metrics.** Fairness definitions vary based on application context. Group fairness, as described in Verma & Rubin (2018), ensures impartiality within demographic groups but may sometimes compromise individual fairness. This trade-off can conflict with

medical ethics, emphasizing the need for methods that prioritize both group and individual fairness Beauchamp (2003). Some argue that Max-Min fairness complements group fairness metrics Zong et al. (2022). Common fairness metrics include Demographic Parity Difference (DPD) Bickel et al. (1975); Agarwal et al. (2018; 2019), Difference in Equal Opportunity (DEO) Hardt et al. (2016), and Difference in Equalized Odds (DEOdds) Agarwal et al. (2018). In crucial medical scenarios, prioritizing group fairness over accuracy is not viable. We introduce an equity scaling mechanism, aligning segmentation accuracy with group fairness, offering a comprehensive perspective on both performance and fairness.

## 3 Fair Disc-Cup Segmentation

**Disc-Cup Segmentation Framework.** In ophthalmology, disc-cup segmentation serves as a foundational step in assessing the optic nerve head structures and diagnosing glaucoma in its early stage. We represent a scanning laser ophthalmoscopy (SLO) fundus image as $x \in \mathcal{X} \subset \mathbb{R}^{H \times W \times C}$, where $H \times W$ denotes the spatial resolution and $C$ signifies the channel count. Our objective is to predict a segmentation map, $\hat{S}$, with a resolution of $H \times W$. Every pixel in this map is associated with a class from the set $\mathcal{Y} = \{y_0, y_1, y_2\}$, where $y_0$ represents the background, $y_1$ denotes the optic disc, and $y_2$ indicates the cup. The overall objective is to construct a model $f_\theta : \mathbb{R}^d \xrightarrow{\theta} \hat{S}$ capable of predicting this segmentation mask $\hat{S}$, where $\theta$ denotes the segmentation model's parameters. The segmentation model $f_\theta$ can be implemented using various SOTA models, such as SAM Kirillov et al. (2023) and TransUNet Chen et al. (2021).

This task is important because vertical cup-to-disc ratio (CDR) Jonas et al. (2000) is accepted and commonly used by clinicians as a prime metric for the early detection of glaucoma. It is computed by determining the ratio of the vertical cup diameter (VCD) to the vertical disc diameter (VDD). An SLO fundus image vividly portrays the optic disc (OD) as a distinguishable bright white/gray oval in its center, split into the inner bright optic cup (OC) and the outer neuroretinal rim. An increasing CDR generally hints at a larger risk of glaucoma, underscoring the importance of precise disc and cup segmentation.

Setting out with an ambition of fair segmentation, our FairSeg framework aims to ensure consistent and unbiased disc-cup delineation across diverse demographic groups. In practice, we observe significant bias between different demographic groups, and this can be caused by many factors (e.g., healthy subjects from the Black racial group tend to obtain slightly larger CDR than other groups). This venture is not merely a technical pursuit but carries profound societal implications, making it crucial to infuse fairness in our methodology.

**Incorporating Fairness in Segmentation.** Attaining accurate segmentations remains pivotal; however, ensuring the system's fairness across various demographic groups also amplifies its clinical importance. Thus, in our Harvard-FairSeg dataset, each image comes with an associated sensitive attribute $a \in \mathcal{A}$. This attribute embodies critical social demographics such as race $\mathcal{A} = \{\text{Asian}, \text{Black or African American}, \text{White or Caucasian}\}$, gender $\mathcal{A} = \{\text{Female}, \text{Male}\}$, ethnicity $\mathcal{A} = \{\text{Non-Hispanic}, \text{Hispanic}\}$, and preferred language $\mathcal{A} = \{\text{English}, \text{Spanish}, \text{Others}\}$. For simplicity, we digitize these categories to $\mathcal{A} = \{0, 1, 2\}$ or $\mathcal{A} = \{0, 1\}$. Incorporating the fairness learning paradigms into disc-cup segmentation, our goal extends to training $f_\theta$ not only for segmentation precision but also to ensure equitable performance across the diverse sensitive attributes.

## 4 Dataset Analysis

**Data Collection and Quality Control.** Our institute's institutional review board (IRB) approved this study, which followed the principles of the Declaration of Helsinki. Since the study was retrospective, the IRB waived the requirement for informed consent from patients.

The subjects tested between 2010 and 2021 are from a large academic eye hospital. There are three types of data to be released in this study: (1) SLO fundus imaging scans; (2) patient demographics; and (3) disc-cup masks annotated automatically from the OCT machine and manually graded by professional medical practitioners. Particularly, the pixel annotations of disc and cup regions are first acquired from the OCT machine, where the disc border in 3D

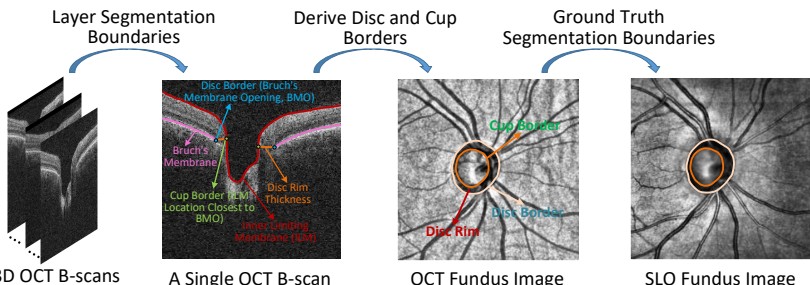

Figure 1: The process to obtain ground truth disc and cup boundaries on the SLO fundus image. The OCT and SLO fundus images have been previously registered using NiftyReg.

OCT is segmented as the Bruch's membrane opening by the OCT manufacturer software, and the cup border is detected as the intersection between the inner limiting membrane (ILM) and the plane that results in minimum surface area from the intersection and disc border on the plane Mitsch et al. (2019); Everett & Oakley (2015). Approximately, cup border can be considered as the closest location on the ILM to the disc border, which is defined as the Bruch's membrane opening. Both Bruch's membrane opening and the internal limiting membrane can be easily segmented due to the high contrast between them and the background. Since the OCT manufacturer software leverages 3D information, the disc and cup segmentation is generally reliable. In comparison, 2D disc and cup segmentation on fundus photos can be challenging due to various factors including attenuated imaging signal and blood vessel obstruction. However, OCT machines are fairly expensive and less prevalent in primary care, therefore, we propose to migrate those annotations from 3D OCT to 2D SLO fundus for potentially broader impact in early-stage glaucoma screening in the primary care domains. Specifically, we first align the SLO fundus images with OCT-derived fundus (OCT fundus) images utilizing the NiftyReg registration tool Modat et al. (2014). Subsequently, the affine metric from NiftyReg is applied to the disc-cup masks of the OCT fundus images, aligning them to the SLO fundus images. This procedure effectively produces a plethora of high-caliber SLO fundus mask annotations, sidestepping the labor-intensive manual pixel annotation process. It's noteworthy that this registration operation demonstrates considerable precision in real-world scenarios, as evident from our empirical observations that highlight an approximate 80% success rate in registrations. Post this automated process, the generated masks undergo rigorous examination and are hand-graded by a panel of five medical professionals to ascertain the precise annotation of the disc-cup regions and exclude masks with incorrect disc or cup locations and failed registrations.

**Data Characteristics.** Our Harvard-FairSeg dataset contains 10,000 samples from 10,000 subjects. We divide our data into the training set with 8,000 samples, and the test set with 2,000 samples. The dataset's collective age average is $60.3 \pm 16.5$ years. Within this dataset, six sensitive attributes including age, gender, race, ethnicity, preferred language, and marital status, are included for in-depth fairness learning studies. In particular, regarding racial demographics, the dataset includes samples from three major groups: Asian, with 919 samples; Black, with 1,473 samples; and White, with 7,608 samples. Gender-wise, females constitute 58.5% of the subjects, with the remainder being males. The ethnic distribution is highlighted by 90.6% Non-Hispanic, 3.7% Hispanic, and 5.7% unspecified. In terms of preferred language, 92.4% of the subjects prefer English, 1.5% prefer Spanish, 1% prefer other languages, and 5.1% remain unidentified. From a marital status perspective, 57.7% are in a marriage or partnered, 27.1% are single, 6.8% have experienced divorce, 0.8% are legally separated, 5.2% are widowed, and 2.4% are not specified.

## 5 Methodology for Understanding Debiasing

In this section, we introduce our proposed fair error-bound scaling approach, which modifies the Dice loss to adjust pixel-wise weights of samples from different sensitive groups. Moreover, after recognizing the potential shortcomings of standard fairness metrics in medical contexts is primarily their potential discord with overall performance (i.e., Dice for segmentation or AUC for classification) and their sensitivity to data quality, we further

advocate for an equity scaling metric for segmentation tasks. This metric integrates demographic-dependent equity considerations with primary segmentation performance metrics, including Dice and IoU.

**Fair Error-Bound Scaling.** As aforementioned, we denote an image $x \in \mathcal{X} \subset \mathbb{R}^{H \times W \times C}$, and our objective is to predict a segmentation map, $\hat{S}$, with a resolution of $H \times W$. Every pixel in this map is associated with a class from the set $\mathcal{Y} = \{y_0, y_1, y_2\}$, where $y_0$ represents the background, $y_1$ denotes the optic disc, and $y_2$ indicates the cup. Each image comes with an associated sensitive attribute $a \in \mathcal{A}$. This attribute embodies critical social demographics that have been introduced in Section. 3. We hypothesize that the group of samples that obtain smaller overall dice loss means the model learns better for the samples from that certain group, and therefore, requires smaller weights for these groups of samples. While, in contrast, the group with larger overall dice loss (i.e., hard cases) may lead to worse generalization and cause more algorithmic bias, which requires larger learning weights for these groups of samples. Therefore, we propose a new fair error-bound scaling method for scaling the dice losses between different demographic groups during training. We first define the standard Dice loss between predicted pixel scores $\hat{y}$ and the ground truth targets $y$ is represented as:

$$D(\hat{y}, y) = 1 - \frac{2 \times \mathcal{I}(\hat{y}, y) + \epsilon}{\sum \hat{y}^2 + \sum y^2) + \epsilon}, \tag{1}$$

where the intersect function $\mathcal{I}$ calculates the element-wise multiplication between the predicted pixel scores $\hat{y}$ and the target $y$.

To ensure fairness across different attribute groups, we augment the Dice loss above with a novel Fair Error-Bound Scaling mechanism. This results in the loss function below:

$$\mathcal{L}_{\text{fair}}(\hat{y}, y, a) = \frac{1}{|\mathcal{Y}|} \sum_{i=1}^{|\mathcal{Y}|} \Omega_i \times D(\hat{y}_i \times \sum_{a=1}^{|\mathcal{A}|} \mathcal{W}_a, y_i), \tag{2}$$

where $\Omega_i$ denotes the weight for target class $i$, and $\mathcal{W}_a$ is the weight list corresponding to a particular attribute. The weight list $\mathcal{W}$ for the sensitive attribute $a \in \mathcal{A}$ can be formulated as:

$$\mathcal{W}_a = \tanh\left(\left(\frac{\min(\sum_{i=1}^{|\mathcal{Y}|} D(\hat{y}[a]_i, y[a]_i))}{\sum_{i=1}^{|\mathcal{Y}|} D(\hat{y}[a]_i, y[a]_i)}\right)^\gamma\right) \tag{3}$$

By modulating predicted pixel scores with these attribute weights, this loss ensures that different attribute groups contribute balancedly to the loss function during model training, thereby promoting fairness.

**Equity-Scaled Metric for Fair Segmentation.** In the field of medical image segmentation, the need to address demographic fairness has become increasingly paramount. Traditional segmentation metrics like Dice and IoU offer insights into the segmentation performance but may not capture the fairness across diverse demographic groups effectively. Considering this, our goal is to present a new metric that encapsulates both the efficiency of the segmentation and its fairness across different demographics. This gives rise to a holistic perspective, ensuring models that are both accurate and fair.

Let $\mathcal{I} \in \{\text{Dice}, \text{IoU}, \dots\}$ represent a segmentation metric. Conventional segmentation metrics, like Dice and IoU, can be expressed as $\mathcal{M}(\{(z', y)\})$, where they assess the overlap between the predicted segmentation $z'$ and the ground truth $y$. However, these metrics often overlook the demographic-sensitive attribute information of the samples. Inspired by Luo et al. (2023c), to incorporate group-wise fairness, we need to evaluate the performance across each demographic group individually. Let's denote the set of all demographic groups as $\mathcal{A}$. To integrate demographic fairness, we first define a performance discrepancy $\Delta$ for segmentation metrics, similar to Equation (4), as follows:

$$\Delta = \sum_{A \in \mathcal{A}} |\mathcal{I}(\{(z', y)\}) - \mathcal{I}(\{(z', a, y)|a = A\})| \tag{4}$$

Here, $\Delta$ measures the aggregate deviation of each demographic group's performance from the overall performance. It approaches zero when all groups achieve similar segmentation accuracy.

Table 2: **Optic Cup and Rim** segmentation performance on the Harvard-FairSeg dataset with **race** as the sensitive attribute.

| | Method | Overall ES-Dice↑ | Overall Dice↑ | Overall ES-IoU↑ | Overall IoU↑ | Asian Dice↑ | Black Dice↑ | White Dice↑ | Asian IoU↑ | Black IoU↑ | White IoU↑ |
|---|---|---|---|---|---|---|---|---|---|---|---|
| Cup | SAMed | 0.8532 | 0.8671 | 0.7643 | 0.7813 | 0.8568 | **0.8730** | 0.8670 | 0.7688 | **0.7905** | 0.7808 |
| | SAMed+ADV | **0.8591** | **0.8698** | **0.7702** | **0.7840** | **0.8590** | 0.8705 | **0.8708** | 0.7709 | 0.7882 | **0.7846** |
| | SAMed+GroupDRO | 0.8582 | 0.8695 | 0.7700 | 0.7838 | 0.8583 | 0.8704 | 0.8706 | **0.7711** | 0.7886 | 0.7842 |
| | **SAMed+FEBS** | 0.8566 | 0.8671 | 0.7671 | 0.7808 | 0.8587 | 0.8708 | 0.8672 | 0.7708 | 0.7882 | 0.7804 |
| | TransUNet | **0.8281** | **0.8481** | **0.7300** | **0.7532** | **0.8270** | 0.8489 | **0.8503** | **0.7277** | **0.7576** | **0.7551** |
| | TransUNet+ADV | 0.8256 | 0.8410 | 0.7265 | 0.7432 | 0.8246 | 0.8417 | 0.8426 | 0.7260 | 0.7482 | 0.7440 |
| | TransUNet+GroupDRO | 0.8201 | 0.8442 | 0.7252 | 0.7479 | 0.8197 | 0.8469 | 0.8464 | 0.7232 | 0.7529 | 0.7495 |
| | **TransUNet+FEBS** | 0.8253 | 0.8464 | 0.7265 | 0.7497 | 0.8248 | **0.8484** | 0.8484 | 0.7247 | 0.7550 | 0.7513 |
| Rim | SAMed | 0.7478 | 0.8291 | 0.6395 | 0.7217 | 0.7890 | 0.7758 | 0.8444 | 0.6743 | 0.6587 | 0.7399 |
| | SAMed+ADV | 0.7394 | 0.8235 | 0.6300 | 0.7138 | 0.7801 | 0.7691 | 0.8395 | 0.6635 | 0.6498 | 0.7325 |
| | SAMed+GroupDRO | 0.7509 | 0.8302 | 0.6427 | 0.7230 | 0.7952 | 0.7748 | 0.8454 | 0.6822 | 0.6568 | 0.7410 |
| | **SAMed+FEBS** | **0.7529** | **0.8323** | **0.6451** | **0.7260** | **0.7952** | **0.7789** | **0.8473** | **0.6825** | **0.6620** | **0.7439** |
| | TransUNet | 0.7034 | 0.7927 | 0.5848 | 0.6706 | 0.7457 | 0.7307 | 0.8106 | 0.6160 | 0.5991 | 0.6913 |
| | TransUNet+ADV | 0.7000 | 0.7906 | 0.5825 | 0.6682 | 0.7413 | 0.7286 | 0.8087 | 0.6116 | 0.5982 | 0.6888 |
| | TransUNet+GroupDRO | 0.7002 | 0.7896 | 0.5814 | 0.6674 | 0.7470 | 0.7229 | 0.8080 | 0.6183 | 0.5899 | 0.6887 |
| | **TransUNet+FEBS** | **0.7050** | **0.7950** | **0.5871** | **0.6725** | **0.7479** | **0.7325** | **0.8130** | **0.6185** | **0.6020** | **0.6935** |

When we consider the fairness across different groups, we need to compute the relative disparity between the overall segmentation accuracy and that within each demographic groups. With this, the Equity-Scaled Segmentation Performance (ESSP) metric, ESSP, is defined as:

$$\text{ESSP} = \frac{\mathcal{I}(\{(z', y)\})}{1 + \Delta} \tag{5}$$

This formulation ensures that ESSP is always less than or equal to $\mathcal{I}$. As $\Delta$ diminishes (indicating equitable segmentation performance across groups), ESSP converges to the traditional metric $\mathcal{I}$. Conversely, a higher $\Delta$ signifies greater disparity in segmentation performance across demographics, resulting in a lower ESSP score. This approach allows us to evaluate segmentation models not only for their accuracy (as measured by Dice, IoU, etc.) but also for their fairness across different demographic groups. This makes the ESSP scoring function a pivotal metric in ensuring both segmentation accuracy and fairness in medical imaging tasks. Please note that an alternative method for gauging equity-scaled segmentation performance involves directly utilizing standard deviation to quantify disparities across different groups. However, this approach, reliant on standard deviation, tends to yield only subtle variations between overall and equity-scaled performances. Such minor discrepancies can restrict its effectiveness in accurately quantifying fairness across various algorithms. For comprehensive insights, we also include results derived from this standard deviation-based metric in the appendix.

# 6 Experiment & Analysis

## 6.1 Algorithms

**Segmentation Models.** We select two segmentation models to investigate the fair segmentation problem, including the recent SOTA SAMed Zhang & Liu (2023) model and a classic TransUNet Chen et al. (2021). The **SAMed** approach Zhang & Liu (2023) offers a novel solution for medical image segmentation, drawing inspiration from the expansive capabilities of the recent Segment Anything Model (SAM) Kirillov et al. (2023). Pioneering a new paradigm, SAMed tailors the large-scale SAM image encoder to cater specifically to medical images. By employing the Low-rank-based (LoRA) fine-tuning strategy, SAMed efficiently adapts the SAM image encoder for medical scenarios. The LoRA modules are integrated into both the prompt encoder and mask decoder, and the model is fine-tuned on labeled medical image segmentation datasets. **TransUNet** Chen et al. (2021) represents a harmonious integration of transformer architectures with the established U-Net design principles, tailored specifically for medical imaging. It combines the strength of CNN backbones for spatial information extraction with the power of transformers to capture long-range dependencies in images. By utilizing the self-attention mechanism of transformers, TransUNet ensures that even distant regions in an image are effectively related, while its U-Net architecture helps preserve vital spatial hierarchies, making it a potent tool for medical image segmentation.

**Fairness Algorithms.** For our proposed fair segmentation, we employed several state-of-the-art fairness techniques as baselines:

Table 3: **Optic Cup and Rim** segmentation performance on the Harvard-FairSeg dataset with **gender** as the sensitive attribute.

| | Method | Overall ES-Dice↑ | Overall Dice↑ | Overall ES-IoU↑ | Overall IoU↑ | Male Dice↑ | Female Dice↑ | Male IoU↑ | Female IoU↑ |
|---|---|---|---|---|---|---|---|---|---|
| Cup | SAMed | 0.8623 | 0.8671 | 0.7757 | 0.7813 | 0.8647 | 0.8703 | 0.7783 | 0.7855 |
| | SAMed+ADV | 0.8655 | 0.8667 | 0.7780 | 0.7803 | 0.8661 | 0.8675 | 0.7791 | 0.7820 |
| | SAMed+GroupDRO | **0.8669** | 0.8671 | **0.7800** | 0.7808 | 0.8672 | 0.8670 | 0.7804 | 0.7814 |
| | **SAMed+FEBS** | 0.8642 | **0.8702** | 0.7758 | **0.7823** | **0.8718** | **0.8756** | **0.7851** | **0.7879** |
| | TransUNet | 0.8435 | 0.8481 | 0.7490 | **0.7532** | 0.8458 | 0.8513 | **0.7508** | **0.7564** |
| | TransUNet+ADV | 0.8342 | 0.8345 | 0.7348 | 0.7356 | 0.8344 | 0.8348 | 0.7361 | 0.7350 |
| | TransUNet+GroupDRO | 0.8405 | 0.8478 | 0.7453 | 0.7522 | 0.8441 | **0.8528** | 0.7483 | 0.7575 |
| | **TransUNet+FEBS** | **0.8464** | **0.8489** | **0.7492** | 0.7530 | **0.8494** | 0.8514 | 0.7505 | 0.7556 |
| Rim | SAMed | 0.8236 | 0.8291 | 0.7158 | 0.7217 | 0.8319 | 0.8252 | 0.7252 | 0.7169 |
| | SAMed+ADV | 0.8244 | 0.8309 | 0.7168 | 0.7236 | 0.8342 | 0.8263 | 0.7276 | 0.7181 |
| | SAMed+GroupDRO | 0.8255 | **0.8320** | 0.7185 | 0.7253 | **0.8353** | 0.8274 | **0.7292** | 0.7198 |
| | **SAMed+FEBS** | **0.8277** | 0.8318 | **0.7216** | 0.7253 | 0.8338 | **0.8289** | 0.7274 | **0.7223** |
| | TransUNet | 0.7882 | **0.7927** | 0.6659 | 0.6706 | **0.7951** | 0.7894 | 0.6736 | 0.6665 |
| | TransUNet+ADV | 0.7754 | 0.7852 | 0.6522 | 0.6630 | 0.7905 | 0.7779 | 0.6699 | 0.6534 |
| | TransUNet+GroupDRO | **0.7893** | 0.7917 | **0.6673** | 0.6699 | 0.7930 | 0.7900 | **0.6716** | **0.6677** |
| | **TransUNet+FEBS** | 0.7851 | 0.7898 | 0.6655 | 0.6698 | 0.7924 | **0.7932** | 0.6678 | 0.6653 |

Table 4: **Optic Cup and Rim** segmentation performance on the Harvard-FairSeg dataset with **preferred language** as the sensitive attribute.

| | Method | Overall ES-Dice↑ | Overall Dice↑ | Overall ES-IoU↑ | Overall IoU↑ | English Dice↑ | Spanish Dice↑ | Others Dice↑ | English IoU↑ | Spanish IoU↑ | Others IoU↑ |
|---|---|---|---|---|---|---|---|---|---|---|---|
| Cup | SAMed | 0.8186 | 0.8671 | 0.7278 | 0.7813 | 0.8652 | 0.9077 | 0.8838 | 0.7791 | 0.8338 | 0.8001 |
| | SAMed+ADV | 0.8197 | 0.8686 | 0.7266 | 0.7830 | 0.8668 | **0.9131** | 0.8820 | 0.7808 | **0.8432** | 0.7982 |
| | SAMed+GroupDRO | 0.8250 | 0.8702 | 0.7329 | **0.7847** | **0.8684** | 0.9085 | **0.8849** | **0.7825** | 0.8360 | **0.8019** |
| | **SAMed+FEBS** | **0.8291** | 0.8684 | **0.7405** | 0.7826 | 0.8670 | 0.9034 | 0.8794 | 0.7810 | 0.8268 | 0.7937 |
| | TransUNet | 0.8037 | **0.8481** | 0.7033 | **0.7532** | 0.8469 | **0.8972** | 0.8531 | **0.7516** | 0.8166 | 0.7592 |
| | TransUNet+ADV | 0.7869 | 0.8312 | 0.6817 | 0.7323 | 0.8296 | 0.8833 | 0.8338 | 0.7301 | 0.7990 | 0.7376 |
| | TransUNet+GroupDRO | 0.7939 | 0.8416 | 0.6932 | 0.7442 | 0.8398 | 0.8844 | **0.8571** | 0.7421 | 0.7993 | 0.7605 |
| | **TransUNet+FEBS** | **0.8040** | **0.8481** | 0.7030 | 0.7523 | 0.8467 | 0.8934 | 0.8562 | 0.7504 | 0.8109 | **0.7619** |
| Rim | SAMed | 0.7852 | 0.8291 | 0.6800 | 0.7217 | 0.8305 | **0.8534** | 0.7989 | 0.7234 | **0.7468** | 0.6871 |
| | SAMed+ADV | 0.7881 | 0.8295 | 0.6823 | 0.7217 | 0.8307 | 0.8528 | 0.8015 | 0.7231 | 0.7463 | 0.6900 |
| | SAMed+GroupDRO | **0.7961** | 0.8311 | **0.6913** | 0.7239 | 0.8322 | 0.8493 | **0.8065** | **0.7253** | 0.7411 | **0.6954** |
| | **SAMed+FEBS** | 0.7892 | **0.8313** | 0.6840 | **0.7244** | **0.8328** | 0.8511 | 0.7992 | 0.7263 | 0.7436 | 0.6865 |
| | TransUNet | 0.7517 | **0.7927** | 0.6346 | **0.6706** | **0.7940** | **0.8165** | 0.7633 | **0.6721** | **0.6950** | 0.6398 |
| | TransUNet+ADV | 0.7521 | 0.7884 | 0.6365 | 0.6666 | 0.7903 | 0.7964 | 0.7501 | 0.6687 | 0.6717 | 0.6265 |
| | TransUNet+GroupDRO | 0.7527 | 0.7857 | 0.6324 | 0.6613 | 0.7867 | 0.8057 | 0.7628 | 0.6625 | 0.6800 | 0.6355 |
| | **TransUNet+FEBS** | **0.7554** | 0.7898 | **0.6379** | 0.6668 | 0.7909 | 0.8106 | **0.7661** | 0.6680 | 0.6865 | **0.6424** |

1. Adversarially Fair Representations (ADV): Introduced by Madras et al. Madras et al. (2018), ADV aims to curate unbiased representations. It refines the model such that sensitive attributes are difficult to deduce from its representations, effectively minimizing inherent biases.

2. Group Distributionally Robust Optimization (GroupDRO): Proposed by Sagawa et al. Sagawa et al. (2019), GroupDRO seeks to enhance model robustness. It minimizes the maximum training loss across all groups by strategically increasing regularization, ensuring the model doesn't exhibit bias towards any particular group.

These fairness algorithms are incorporated into segmentation methods, specifically SAMed and TransUNet, to assess their capability in enhancing fairness and reducing bias across diverse demographic groups during segmentation.

## 6.2 Training Setup

**Datasets**. As we introduce in Section 4, 8,000 SLO fundus images are used for training, and 2,000 SLO fundus images are used for evaluation, with the patient-level split. Apart from the pixel-wise annotation for disc and cup regions, each sample also includes six different sensitive attributes. In this paper, we select four sensitive attributes that obtain the largest group-wise discrepancy in practice, including Race, Gender, Language, and Ethnicity.

**Training and Implementation Details**. For both SAMed and TransUNet, a combined loss function comprising both cross entropy and dice losses is employed as the training loss. The loss function can be defined as

$$L = \lambda_1 \text{CE}(\hat{S}_l, D(S)) + \lambda_2 \text{Dice}(\hat{S}_l, D(S)). \tag{6}$$

Here, CE and Dice stand for the cross entropy loss and Dice loss, respectively. The loss weights $\lambda_1$ and $\lambda_2$ help alternate a balance between the two loss terms. In the initial stages of training, we leverage a warmup strategy to stabilize the model training with our Harvard-FairSeg data. We use the AdamW optimizer with exponential learning rate decay. For the backbone of SAMed and TransUNet, we select the standard ViT-B architecture. We use a learning rate of 0.01, momentum of 0.9, and weight decay 1e-4 for TransUNet. For SAMed, we set the learning rate to 0.005, with momentum and weight decay set to 0.9 and 0.1,

Table 5: **Optic Cup and Rim** segmentation performance on the Harvard-FairSeg dataset with **ethnicity** as the sensitive attribute.

| | Method | Overall ES-Dice↑ | Overall Dice↑ | Overall ES-IoU↑ | Overall IoU↑ | Hispanic Dice↑ | Non-Hispanic Dice↑ | Hispanic IoU↑ | Non-Hispanic IoU↑ |
|---|---|---|---|---|---|---|---|---|---|
| Cup | SAMed | 0.8459 | 0.8671 | 0.7578 | 0.7813 | 0.8653 | **0.8904** | 0.7790 | **0.8100** |
| | SAMed+ADV | 0.8490 | 0.8678 | 0.7595 | 0.7814 | 0.8661 | 0.8883 | 0.7791 | 0.8080 |
| | SAMed+GroupDRO | 0.8550 | 0.8698 | 0.7667 | 0.7840 | 0.8682 | 0.8855 | 0.7819 | 0.8044 |
| | **SAMed+FEBS** | **0.8550** | **0.8685** | **0.7628** | **0.7845** | **0.8704** | 0.8824 | **0.7904** | 0.8070 |
| | TransUNet | 0.8281 | 0.8481 | 0.7300 | 0.7532 | 0.8463 | **0.8704** | 0.7508 | **0.7826** |
| | TransUNet+ADV | 0.8112 | 0.8320 | 0.7083 | 0.7315 | 0.8304 | 0.8561 | 0.7294 | 0.7622 |
| | TransUNet+GroupDRO | **0.8332** | 0.8482 | 0.7358 | 0.7526 | 0.8468 | 0.8648 | 0.7507 | 0.7735 |
| | **TransUNet+FEBS** | 0.8320 | **0.8483** | **0.7359** | **0.7542** | **0.8501** | 0.8661 | **0.7515** | 0.7764 |
| Rim | SAMed | 0.8193 | 0.8291 | 0.7143 | 0.7217 | 0.8277 | 0.8397 | 0.7203 | 0.7307 |
| | SAMed+ADV | 0.8234 | 0.8323 | **0.7184** | **0.7257** | 0.8308 | **0.8416** | 0.7241 | **0.7342** |
| | SAMed+GroupDRO | 0.8212 | 0.8299 | 0.7160 | 0.7224 | 0.8284 | 0.8390 | 0.7208 | 0.7298 |
| | **SAMed+FEBS** | **0.8253** | **0.8331** | 0.7154 | 0.7242 | **0.8349** | 0.8408 | **0.7278** | 0.7329 |
| | TransUNet | 0.7815 | 0.7927 | 0.6626 | 0.6706 | 0.7914 | **0.8057** | 0.6695 | **0.6815** |
| | TransUNet+ADV | 0.7774 | 0.7841 | 0.6557 | 0.6602 | 0.7829 | 0.7915 | 0.6590 | 0.6658 |
| | TransUNet+GroupDRO | 0.7904 | 0.7943 | 0.6672 | 0.6733 | 0.7936 | 0.7901 | **0.6728** | 0.6646 |
| | **TransUNet+FEBS** | **0.7857** | **0.7939** | **0.6692** | **0.6754** | **0.7943** | 0.8040 | 0.6697 | 0.6789 |

respectively. The SAMed model is trained for a total of 130 epochs with an early stopping at the 70 epoch. The TransUNet is trained for 300 epochs without early stopping, following their original implementation. We use a batch size of 42 for training both models. When combining with the fairness algorithms, we use the same setup as aforementioned. For our proposed fair error-bound scaling, we set $\gamma = 1.0$ for all our experiments.

### 6.3 Segmentation & Fairness Evaluation

**Race.** **Table 6** shows the segmentation performance with respect to race, particularly among Asian, Black, and White groups, suggesting that SAMed+ADV, SAMed+GroupDRO, and our method yield relatively comparable results. SAMed+ADV slightly outperforms others in terms of Dice scores among the White subgroup and the overall ES-Dice and ES-IoU. For the Rim, our proposed SAMed+FEBS achieves the best ES-Dice of 0.7529 and the best ES-IoU of 0.6451, which surpasses other methods based on SAMed. TransUNet-based methods achieve relatively worse results than SAMed-based approaches. Notably, our methods with TransUNet as the backbone can surpass other TransUNet-based approaches in terms of ES-Dice and ES-IoU.

**Gender.** When gender serves as the sensitive attribute, the distinctions between methods become subtler, as shown in **Table 3**. Our method combined with the SAMed and TransUNet model consistently outperforms the baseline and its counterpart in terms of ES-Dice and ES-IoU on both Cup and Rim. This showcases the efficacy of our technique in handling gender disparities. Similar to race, TransUNet-based methods also perform worse than the SAMed-based approaches.

**Language.** For results with language attribute in **Table 8**, SAMed+ADV achieves the best performance and fairness for the Cup and rim segmentation when compared with methods based on SAMed. However, the performance of our approaches achieves the best based on TransUNet in terms of ES-Dice and ES-IoU. The disparity between different language groups is the second largest when compared to that of gender and ethnicity.

**Ethnicity.** When considering ethnicity as the sensitive attribute, our method combined with both two segmentation models surpasses the rest in terms of ES-Dice and ES-IoU scores. The details of results with ethnicity are shown in **Table 9**. The performance gaps between Hispanics and Non-Hispanics are larger than gender.

## 7 Conclusion

In this paper, we propose the first dataset to study fairness in medical segmentation. The proposed FairSeg benchmarks are equipped with many SOTA fairness algorithms using different segmentation backbones. Our innovative fair error-bound scaling technique introduces a new fairness dice loss for segmentation tasks. Furthermore, a new equity-scaled evaluation metric provides novel scoring functions for evaluating the fairness of segmentation between different demographic groups. We hope that our contributions not only stimulate further research in this new domain but also serve as a catalyst for the adoption of fairness considerations in medical AI applications universally.

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

# A APPENDIX

**Equity-Scaled Metric Using Standard Deviation.** A modification to the original equity-scaled metric involves calculating the standard deviation Stdev of the segmentation metric for various demographic groups. Let $\mathcal{I} \in \{\text{Dice}, \text{IoU}, \ldots\}$ represent a segmentation metric. Conventional segmentation metrics, like Dice and IoU, can be expressed as $\mathcal{M}(\{(z', y)\})$, where they assess the overlap between the predicted segmentation $z'$ and the ground truth $y$. Let's denote the set of all demographic groups as $\mathcal{A}$.

Given this, the Equity-Scaled Segmentation Performance with standard deviation (ESSP-Stdev) metric for segmentation, ESSP-Stdev, is defined as:

$$\text{ESSP-Stdev} = \frac{\mathcal{I}(\{(z', y)\})}{1 + \text{Stdev}} \tag{7}$$

The denominator in Equation (7) naturally penalizes variations in the segmentation metric across demographics. As Stdev approaches zero, indicating uniform performance across groups, ESSP converges to $\mathcal{I}$. Conversely, as the variability (Stdev) increases, ESSP will diminish, thereby reflecting a penalty for demographic disparity.

A potential limitation of this metric is that it may yield only marginal differences between Equity-Scaled performance and the overall segmentation performance. This subtle distinction could pose a challenge in discerning the effectiveness of various de-biasing approaches in enhancing algorithmic fairness. Nonetheless, for reference purposes, we include results derived from such metrics in this appendix.

Table 6: **Optic Cup and Rim** segmentation performance on the Harvard-FairSeg dataset with **race** as the sensitive attribute.

| | Method | Overall ES-Dice↑ | Overall Dice↑ | Overall ES-IoU↑ | Overall IoU↑ | Asian Dice↑ | Black Dice↑ | White Dice↑ | Asian IoU↑ | Black IoU↑ | White IoU↑ |
|---|---|---|---|---|---|---|---|---|---|---|---|
| Cup | nnUNet | 0.8625 | 0.8710 | 0.7704 | 0.7865 | 0.8675 | 0.8855 | 0.8697 | 0.7639 | 0.8035 | 0.7725 |
| | SAMed | 0.8600 | 0.8671 | 0.7729 | 0.7813 | 0.8568 | **0.8730** | 0.8670 | 0.7688 | 0.7905 | 0.7808 |
| | SAMed+ADV | **0.8640** | **0.8698** | **0.7769** | **0.7840** | **0.8590** | 0.8705 | **0.8708** | 0.7709 | 0.7882 | **0.7846** |
| | SAMed+GroupDRO | 0.8634 | 0.8695 | 0.7767 | 0.7838 | 0.8583 | 0.8704 | 0.8706 | **0.7711** | **0.7886** | 0.7842 |
| | **SAMed+FEBS** | 0.8617 | 0.8671 | 0.7741 | 0.7808 | 0.8587 | 0.8708 | 0.8672 | 0.7708 | 0.7882 | 0.7804 |
| | TransUNet | **0.8372** | **0.8481** | **0.7409** | **0.7532** | **0.8270** | **0.8489** | **0.8503** | **0.7277** | **0.7576** | **0.7551** |
| | TransUNet+ADV | 0.8325 | 0.8410 | 0.7345 | 0.7432 | 0.8246 | 0.8417 | 0.8426 | 0.7260 | 0.7482 | 0.7440 |
| | TransUNet+GroupDRO | 0.8313 | 0.8442 | 0.7359 | 0.7479 | 0.8197 | 0.8469 | 0.8464 | 0.7232 | 0.7529 | 0.7495 |
| | **TransUNet+FEBS** | 0.8350 | 0.8464 | 0.7374 | 0.7497 | 0.8248 | 0.8484 | 0.8484 | 0.7247 | 0.7550 | 0.7513 |
| Rim | nnUNet | 0.8003 | 0.8335 | 0.6959 | 0.7231 | 0.7930 | 0.7682 | 0.8490 | 0.6854 | 0.6639 | 0.7397 |
| | SAMed | 0.8000 | 0.8291 | 0.6919 | 0.7217 | 0.7890 | 0.7758 | 0.8444 | 0.6743 | 0.6587 | 0.7399 |
| | SAMed+ADV | 0.7935 | 0.8235 | 0.6835 | 0.7138 | 0.7801 | 0.7691 | 0.8395 | 0.6635 | 0.6498 | 0.7325 |
| | SAMed+GroupDRO | 0.8011 | 0.8302 | 0.6930 | 0.7230 | 0.7952 | 0.7748 | 0.8454 | 0.6822 | 0.6568 | 0.7410 |
| | **SAMed+FEBS** | **0.8036** | **0.8323** | **0.6963** | **0.7260** | **0.7952** | **0.7789** | **0.8473** | **0.6825** | **0.6620** | **0.7439** |
| | TransUNet | 0.7604 | 0.7927 | 0.6393 | 0.6706 | 0.7457 | 0.7307 | 0.8106 | 0.6160 | 0.5991 | 0.6913 |
| | TransUNet+ADV | 0.7579 | 0.7906 | 0.6371 | 0.6682 | 0.7413 | 0.7286 | 0.8087 | 0.6116 | 0.5982 | 0.6888 |
| | TransUNet+GroupDRO | 0.7564 | 0.7896 | 0.6351 | 0.6674 | 0.7470 | 0.7229 | 0.8080 | 0.6183 | 0.5899 | 0.6887 |
| | **TransUNet+FEBS** | **0.7628** | **0.7950** | **0.6410** | **0.6725** | **0.7479** | **0.7325** | **0.8130** | **0.6185** | **0.6020** | **0.6935** |

**Expansion of Evaluation Metrics.** To provide a more thorough assessment of segmentation accuracy, in Table. 10, we provide the segmentation performances on our Harvard-FairSeg dataset with race as the sensitive attribute in terms of Hausdorff Distance (HD95), Average Surface Distance (ASD), and Normalized Surface Distance (NSD).

Table 7: **Optic Cup and Rim** segmentation performance on the Harvard-FairSeg dataset with **gender** as the sensitive attribute.

| | Method | Overall ES-Dice↑ | Overall Dice↑ | Overall ES-IoU↑ | Overall IoU↑ | Male Dice↑ | Female Dice↑ | Male IoU↑ | Female IoU↑ |
|---|---|---|---|---|---|---|---|---|---|
| Cup | SAMed | 0.8637 | 0.8671 | 0.7773 | 0.7813 | 0.8647 | 0.8703 | 0.7783 | 0.7855 |
| | SAMed+ADV | 0.8658 | 0.8667 | 0.7787 | 0.7803 | 0.8661 | 0.8675 | 0.7791 | 0.7820 |
| | SAMed+GroupDRO | 0.8670 | 0.8671 | 0.7803 | 0.7808 | 0.8672 | 0.8670 | 0.7804 | 0.7814 |
| | **SAMed+FEBS** | **0.8678** | **0.8702** | **0.7807** | **0.7823** | **0.8718** | **0.8756** | **0.7851** | **0.7879** |
| | TransUNet | 0.8448 | 0.8481 | 0.7502 | 0.7532 | 0.8458 | 0.8513 | 0.7508 | 0.7564 |
| | TransUNet+ADV | 0.8343 | 0.8345 | 0.7351 | 0.7356 | 0.8344 | 0.8348 | 0.7361 | 0.7350 |
| | TransUNet+GroupDRO | 0.8426 | 0.8478 | 0.7473 | 0.7522 | 0.8441 | 0.8528 | 0.7483 | 0.7575 |
| | **TransUNet+FEBS** | **0.8477** | **0.8489** | **0.7502** | **0.7530** | **0.8494** | **0.8514** | **0.7505** | **0.7556** |
| Rim | SAMed | 0.8251 | 0.8291 | 0.7175 | 0.7217 | 0.8319 | 0.8252 | 0.7252 | 0.7169 |
| | SAMed+ADV | 0.8263 | 0.8309 | 0.7188 | 0.7236 | 0.8342 | 0.8263 | 0.7276 | 0.7181 |
| | SAMed+GroupDRO | 0.8274 | 0.8320 | 0.7205 | 0.7253 | **0.8353** | 0.8274 | **0.7292** | 0.7198 |
| | **SAMed+FEBS** | **0.8289** | **0.8318** | **0.7227** | **0.7253** | 0.8338 | **0.8289** | 0.7274 | **0.7223** |
| | TransUNet | 0.7895 | 0.7927 | 0.6673 | 0.6706 | 0.7951 | 0.7894 | **0.6736** | **0.6665** |
| | TransUNet+ADV | 0.7783 | 0.7852 | 0.6553 | 0.6630 | 0.7905 | 0.7779 | 0.6699 | 0.6534 |
| | TransUNet+GroupDRO | **0.7901** | **0.7917** | 0.6681 | 0.6699 | **0.7930** | 0.7900 | 0.6716 | 0.6677 |
| | **TransUNet+FEBS** | 0.7893 | 0.7898 | **0.6698** | **0.6698** | 0.7924 | **0.7932** | 0.6678 | 0.6653 |

Table 8: **Optic Cup and Rim** segmentation performance on the Harvard-FairSeg dataset with **preferred language** as the sensitive attribute.

| | Method | Overall ES-Dice↑ | Overall Dice↑ | Overall ES-IoU↑ | Overall IoU↑ | English Dice↑ | Spanish Dice↑ | Others Dice↑ | English IoU↑ | Spanish IoU↑ | Others IoU↑ |
|---|---|---|---|---|---|---|---|---|---|---|---|
| Cup | SAMed | 0.8490 | 0.8671 | 0.7603 | 0.7813 | 0.8652 | 0.9077 | 0.8838 | 0.7791 | 0.8338 | 0.8001 |
| | SAMed+ADV | 0.8485 | 0.8686 | 0.7586 | 0.7830 | 0.8668 | **0.9131** | 0.8820 | 0.7808 | **0.8432** | 0.7982 |
| | SAMed+GroupDRO | **0.8530** | **0.8702** | 0.7640 | **0.7847** | **0.8684** | 0.9085 | **0.8849** | **0.7825** | 0.8360 | **0.8019** |
| | **SAMed+FEBS** | 0.8527 | 0.8684 | **0.7646** | 0.7826 | 0.8670 | 0.9034 | 0.8794 | 0.7810 | 0.8268 | 0.7937 |
| | TransUNet | 0.8255 | **0.8481** | 0.7273 | **0.7532** | **0.8469** | **0.8972** | 0.8531 | **0.7516** | **0.8166** | 0.7592 |
| | TransUNet+ADV | 0.8071 | 0.8312 | 0.7056 | 0.7323 | 0.8296 | 0.8833 | 0.8338 | 0.7301 | 0.7990 | 0.7376 |
| | TransUNet+GroupDRO | 0.8231 | 0.8416 | 0.7231 | 0.7442 | 0.8398 | 0.8844 | **0.8571** | 0.7421 | 0.7993 | 0.7605 |
| | **TransUNet+FEBS** | **0.8277** | **0.8481** | **0.7289** | 0.7523 | 0.8467 | 0.8934 | 0.8562 | 0.7504 | 0.8109 | **0.7619** |
| Rim | SAMed | 0.8070 | 0.8291 | 0.7006 | 0.7217 | 0.8305 | **0.8534** | 0.7989 | 0.7234 | **0.7468** | 0.6871 |
| | SAMed+ADV | 0.8087 | 0.8295 | 0.7019 | 0.7217 | 0.8307 | 0.8528 | 0.8015 | 0.7231 | 0.7463 | 0.6900 |
| | SAMed+GroupDRO | **0.8136** | 0.8311 | **0.7075** | 0.7239 | 0.8322 | 0.8493 | **0.8065** | 0.7253 | 0.7411 | **0.6954** |
| | **SAMed+FEBS** | 0.8100 | **0.8313** | 0.7038 | 0.7244 | **0.8328** | 0.8511 | 0.7992 | **0.7263** | 0.7436 | 0.6865 |
| | TransUNet | 0.7721 | **0.7927** | **0.6525** | **0.6706** | **0.7940** | **0.8165** | 0.7633 | **0.6721** | **0.6950** | 0.6398 |
| | TransUNet+ADV | 0.7690 | 0.7884 | 0.6501 | 0.6666 | 0.7903 | 0.7964 | 0.7501 | **0.6687** | 0.6717 | 0.6265 |
| | TransUNet+GroupDRO | 0.7691 | 0.7857 | 0.6468 | 0.6613 | 0.7867 | 0.8057 | 0.7628 | 0.6625 | 0.6800 | 0.6355 |
| | **TransUNet+FEBS** | **0.7725** | 0.7898 | 0.6524 | 0.6668 | 0.7909 | 0.8106 | **0.7661** | 0.6680 | 0.6865 | **0.6424** |

Table 9: **Optic Cup and Rim** segmentation performance on the Harvard-FairSeg dataset with **ethnicity** as the sensitive attribute.

| | Method | Overall ES-Dice↑ | Overall Dice↑ | Overall ES-IoU↑ | Overall IoU↑ | Hispanic Dice↑ | Non-Hispanic Dice↑ | Hispanic IoU↑ | Non-Hispanic IoU↑ |
|---|---|---|---|---|---|---|---|---|---|
| Cup | SAMed | 0.8519 | 0.8671 | 0.7645 | 0.7813 | 0.8653 | 0.8904 | 0.7790 | **0.8100** |
| | SAMed+ADV | 0.8544 | 0.8678 | 0.7657 | 0.7814 | 0.8661 | **0.8883** | 0.7791 | 0.8080 |
| | SAMed+GroupDRO | 0.8594 | **0.8698** | 0.7718 | 0.7840 | 0.8682 | 0.8855 | 0.7819 | 0.8044 |
| | **SAMed+FEBS** | **0.8611** | 0.8685 | **0.7753** | **0.7845** | **0.8704** | 0.8824 | **0.7904** | 0.8070 |
| | TransUNet | 0.8339 | 0.8481 | 0.7366 | 0.7532 | 0.8463 | **0.8704** | 0.7508 | **0.7826** |
| | TransUNet+ADV | 0.8171 | 0.8320 | 0.7149 | 0.7315 | 0.8304 | 0.8561 | 0.7294 | 0.7622 |
| | TransUNet+GroupDRO | 0.8376 | 0.8482 | 0.7406 | 0.7526 | 0.8468 | 0.8648 | 0.7507 | 0.7735 |
| | **TransUNet+FEBS** | **0.8388** | **0.8483** | **0.7412** | **0.7542** | **0.8501** | 0.8661 | **0.7515** | 0.7764 |
| Rim | SAMed | 0.8221 | 0.8291 | 0.7164 | 0.7217 | 0.8277 | 0.8397 | 0.7203 | 0.7307 |
| | SAMed+ADV | 0.8260 | 0.8323 | 0.7206 | **0.7257** | 0.8308 | **0.8416** | 0.7241 | **0.7342** |
| | SAMed+GroupDRO | 0.8237 | 0.8299 | 0.7178 | 0.7224 | 0.8284 | 0.8390 | 0.7208 | 0.7298 |
| | **SAMed+FEBS** | **0.8296** | **0.8331** | **0.7215** | 0.7242 | **0.8349** | 0.8408 | **0.7278** | 0.7329 |
| | TransUNet | 0.7848 | 0.7927 | 0.6650 | 0.6706 | 0.7914 | **0.8057** | 0.6695 | **0.6815** |
| | TransUNet+ADV | 0.7793 | 0.7841 | 0.6570 | 0.6602 | 0.7829 | 0.7915 | 0.6590 | 0.6658 |
| | TransUNet+GroupDRO | **0.7924** | **0.7943** | 0.6694 | 0.6733 | 0.7936 | 0.7901 | **0.6728** | 0.6646 |
| | **TransUNet+FEBS** | 0.7884 | 0.7939 | **0.6710** | **0.6754** | **0.7943** | 0.8040 | 0.6697 | 0.6789 |

Table 10: **Optic Cup and Rim** HD95, ASD, and NSD performances on the Harvard-FairSeg dataset with **race** as the sensitive attribute.

| | Method | Overall HD95↓ | Asian HD95↓ | Black HD95↓ | White HD95↓ | Overall ASD↓ | Asian ASD↓ | Black ASD↓ | White ASD↓ | Overall NSD↑ | Asian NSD↑ | Black NSD↑ | White NSD↑ |
|---|---|---|---|---|---|---|---|---|---|---|---|---|---|
| Cup | SAMed | 9.6231 | 11.0005 | 11.0142 | 9.1866 | 3.9650 | 4.6765 | 4.7743 | 3.7209 | 0.7222 | 0.6825 | 0.6871 | 0.7338 |
| | SAMed+ADV | 9.4594 | 11.0170 | 11.1193 | 8.9479 | 4.0123 | 4.8623 | 4.9236 | 3.7321 | 0.7405 | 0.7045 | 0.6954 | 0.7537 |
| | SAMed+GroupDRO | 9.4633 | 11.0891 | 11.0449 | 8.9603 | 3.9494 | 4.6462 | 4.8634 | 3.6855 | 0.7415 | 0.7093 | 0.6942 | 0.7547 |
| | **SAMed+FEBS** | 9.4494 | 10.8376 | 11.1637 | 8.9453 | 3.9288 | 4.7042 | 4.8210 | 3.6606 | 0.7399 | 0.7038 | 0.7015 | 0.7518 |
| | TransUNet | 4.7577 | 5.6983 | 5.5411 | 4.4937 | 2.0552 | 2.3692 | 2.4573 | 1.9382 | 0.9314 | 0.8845 | 0.8908 | 0.9449 |
| | TransUNet+ADV | 5.0157 | 5.9379 | 5.8292 | 4.7476 | 2.1319 | 2.4693 | 2.4978 | 2.0198 | 0.9208 | 0.8819 | 0.8817 | 0.9331 |
| | TransUNet+GroupDRO | 4.8195 | 5.6424 | 5.6753 | 4.5535 | 2.0615 | 2.3637 | 2.4952 | 1.9393 | 0.9285 | 0.8768 | 0.8873 | 0.9426 |
| | **TransUNet+FEBS** | 4.9603 | 5.8818 | 5.7398 | 4.6992 | 2.1441 | 2.4641 | 2.5445 | 2.0268 | 0.9262 | 0.8796 | 0.8850 | 0.9397 |
| Rim | SAMed | 9.9379 | 11.4859 | 11.5671 | 9.4337 | 3.9353 | 4.5013 | 4.5473 | 3.7476 | 0.7483 | 0.7313 | 0.7215 | 0.7556 |
| | SAMed+ADV | 8.8316 | 10.5364 | 10.2364 | 8.3562 | 3.3523 | 3.9946 | 3.8975 | 3.1700 | 0.8063 | 0.7694 | 0.7686 | 0.8182 |
| | SAMed+GroupDRO | 8.7609 | 10.4638 | 10.0592 | 8.3076 | 3.3473 | 3.8345 | 3.9499 | 3.1702 | 0.8078 | 0.7760 | 0.7696 | 0.8191 |
| | **SAMed+FEBS** | 8.7930 | 10.3167 | 10.3407 | 8.3082 | 3.4174 | 4.0053 | 4.0608 | 3.2208 | 0.8043 | 0.7732 | 0.7704 | 0.8146 |
| | TransUNet | 4.4404 | 5.4496 | 5.0893 | 4.1964 | 1.7534 | 1.9643 | 1.9941 | 1.6809 | 0.9601 | 0.9326 | 0.9326 | 0.9688 |
| | TransUNet+ADV | 4.6301 | 5.6065 | 5.4390 | 4.3569 | 1.8110 | 2.1085 | 2.0885 | 1.7213 | 0.9554 | 0.9245 | 0.9221 | 0.9656 |
| | TransUNet+GroupDRO | 4.5268 | 5.4243 | 5.2290 | 4.2842 | 1.7498 | 1.9683 | 2.0017 | 1.6742 | 0.9596 | 0.9322 | 0.9307 | 0.9685 |
| | **TransUNet+FEBS** | 4.5311 | 5.5251 | 5.2808 | 4.2682 | 1.8321 | 2.0642 | 2.0771 | 1.7564 | 0.9581 | 0.9306 | 0.9315 | 0.9666 |

