# OpenReview forum: "FairSeg: A Large-Scale Medical Image Segmentation Dataset for Fairness Learning Using Segment Anything Model with Fair Error-Bound Scaling"
_ICLR.cc/2024/Conference — ICLR 2024 poster_

### Official Review · Reviewer_f8dv · 2023-10-17

**Soundness:** 4 excellent
**Presentation:** 4 excellent
**Contribution:** 3 good
**Rating:** 6
**Confidence:** 5

**Summary:**

This paper proposed a dataset for retinal disc/cup segmentation with several pre-defined attributes, which should be useful for studying the fairness problem in the medical domain. Furthermore, the authors set a baseline for the problem and define the evaluation metrics in this scenario. Overall, this work is sound and meaningful.

**Strengths:**

[1] Providing a dataset for fairness-related research is meaningful for the current community, along with its baseline and evaluation setting.

[2] Good writing and clear motivation

**Weaknesses:**

[1] ICLR might not be the best place for this paper. Other medical journals or conferences would be more suitable.

[2] There are many evaluation ways to assess the fairness problem. The selected metrics might not be the most suitable one. Please elaborate more on the motivation of baseline setting and evaluation.

[3] Some current works should be included to make the experiments sufficient. See: FairAdaBN: Mitigating unfairness with adaptive batch normalization and its application to dermatological disease classification

[4] Since most of the attributes are only for the patient level, why use the pixel-wise weights?

**Questions:**

See the above weaknesses

**Details Of Ethics Concerns:**

This work proposed a retinal dataset with several attributes, which should be further checked from the ethics view.

---

> ### Author Response · Authors · 2023-11-22
> **Response to Reviewer f8dv**
>
> Thank you so much for your review and the insightful comments.
>
>
> **ICLR might not be the best place for this paper. Other medical journals or conferences would be more suitable.**
>
> Our primary contribution centers on fairness in machine learning, rather than on medical imaging itself. We regard medical imaging as a crucial area for studying fairness, given its human-centric nature and the potentially severe consequences of unfairness in medical deep learning systems in real-world scenarios.
>
>
> **There are many evaluation ways to assess the fairness problem. The selected metrics might not be the most suitable one. Please elaborate more on the motivation of baseline setting and evaluation.**
>
>
> We introduced a novel metric for assessing fairness in medical segmentation, called Equity-Scaled Segmentation Performance (ESSP). ESSP offers a more direct and clinician-friendly evaluation compared to traditional fairness metrics like DPD and DEOdds. Unlike DPD and DEOdds, which may overlook overall performance, leading to a situation where a model with uniformly lower performance across all demographics could falsely appear fairer. This misalignment is particularly problematic in safety-critical medical applications, which demand high accuracy.  In contrast, our proposed ESSP addresses this issue. It evaluates not only the disparity across different demographic groups but also measures the extent to which overall segmentation accuracy is compromised to achieve a fairer model. A higher ESSP score indicates a model that achieves both fairness and high accuracy.
>
> Alongside ESSP, according to the reviewer’s suggestion, we have also updated the DPD and DEOdds results in the supplementary material, providing a comprehensive fairness assessment of various segmentation approaches. As the table below illustrates, our FEBS (Fairness-Enhanced Biased Segmentation) approach outperforms previous methods in terms of DPD and DEOdds, highlighting its effectiveness in achieving fairer outcomes in medical imaging. From the table below, we show that our method achieves comparable DPD and DEodds performances when compared against other segmentation methods.
>
> |      | Method            | Overall DPD | Overall DEodds |
> |------|-------------------|-------------|----------------|
> | Cup  | SAMed             | 0.0085      | 0.0196         |
> |      | SAMed+ADV         | 0.0079      | 0.0071         |
> |      | SAMed+GroupDRO    | 0.0078      | 0.0154         |
> |      | Ours (SAMed)      | 0.0079      | 0.0020         |
> |      | TransUNet         | 0.0083      | 0.0430         |
> |      | TransUNet+ADV     | 0.0074      | 0.0389         |
> |      | TransUNet+GroupDRO| 0.0081      | 0.0317         |
> |      | Ours (TransUNet)  | 0.0085      | 0.0492         |
> | Rim  | SAMed             | 0.0005      | 0.0670         |
> |      | SAMed+ADV         | 0.0004      | 0.0657         |
> |      | SAMed+GroupDRO    | 0.0009      | 0.0792         |
> |      | Ours (SAMed)      | 0.0002      | 0.0715         |
> |      | TransUNet         | 0.0014      | 0.0877         |
> |      | TransUNet+ADV     | 0.0025      | 0.0708         |
> |      | TransUNet+GroupDRO| 0.0018      | 0.0699         |
> |      | Ours (TransUNet)  | 0.0014      | 0.0822         |
>
>
>
> **Some current works should be included to make the experiments sufficient. See: FairAdaBN: Mitigating unfairness with adaptive batch normalization and its application to dermatological disease classification.**
>
>
>
> FairAdaBN was originally proposed for classification tasks, not segmentation, and necessitated modifications to the backbone architectures of classification models. However, with the advent of recent large-scale segmentation models like Meta's SAM, these models require fine-tuning with pre-trained weights for optimal performance. Altering the architecture, as in the case of FairAdaBN, might hinder the loading of these pre-trained parameters, which are trained from extensive datasets, potentially leading to lower segmentation accuracy. Additionally, FairAdaBN was initially applied to a ResNet, and adapting it to Transformer-based segmentation models also presents challenges. Thus, we suggest that further investigation is needed before applying FairAdaBN to segmentation tasks. We have cited and discussed FairAdaBN in our related work section, and we aim to integrate the FairAdaBN approach into our future research.
>
>
> **Since most of the attributes are only for the patient level, why use the pixel-wise weights?**
>
> Patients from various demographic groups may exhibit different anatomical characteristics. For example, Black people often have a larger cup-to-disc ratio and cup area compared to other races. These anatomical differences can influence segmentation accuracy. To address this, we employ pixel-wise weights to accommodate these underlying anatomical variations within the fundus images.

---

### Official Review · Reviewer_b2i2 · 2023-10-31

**Soundness:** 4 excellent
**Presentation:** 4 excellent
**Contribution:** 3 good
**Rating:** 8
**Confidence:** 4

**Summary:**

This paper proposed a fundus image dataset for benchmarking the fairness of medical image segmentation methods, which is the first dataset and benchmark in this field. The authors also proposed to rescale the loss function with the upper training error-bound of each identity group to tackle the fairness issue.

**Strengths:**

- Novel Dataset: The paper introduced FairSeg, a new dataset for medical image segmentation with a focus on fairness. The creation of such a dataset is valuable as it addresses a gap in the current availability of medical datasets with fairness considerations.

- Fairness-Oriented Methodology: The authors proposed a fair error-bound scaling approach and an equity scaling metric. These methods represent an advanced effort to integrate fairness directly into the model training process, which could lead to more equitable healthcare outcomes.

- Open Access: I like that the author released the dataset and code for reproducibility and further research, which is a strong aspect of this work.

**Weaknesses:**

- Dice and IoU are equivalent (https://www.sciencedirect.com/science/article/pii/S1361841521000815), which are not necessary to be reported simultaneously. Instead, please add NSD which is suggested by metrics reloaded (https://arxiv.org/abs/2206.01653).

- nnUNet is still the state-of-the-art in many segmentation tasks. It would be great to evaluate it on your dataset.

**Questions:**

- The dataset was released as npz format. Could you please also release the original format?

- It would be great if you could release the trained models as well.

- Where do you plan to host this benchmark? CodaLab could be a good platform.

---

> ### Author Response · Authors · 2023-11-22
> **Response to Reviewer b2i2**
>
> We thank the reviewer for the positive and encouraging review.
>
> **Please add NSD which is suggested by metrics reloaded.**
>
> Dice and IoU are commonly used by previous research [1,2,3]. For the sake of future comparison and completeness, we will keep the Dice and IoU in the table, and add NSD as the additional metric. Furthermore, we have included the NSD results for race as table below. The table indicates that unfairness commonly exists in our proposed fair segmentation task, which further strengthens the contribution of our FairSeg dataset. We have added those results in the supplementary material.
>
> |      |                       | **Overall NSD↑** | **Asian NSD↑** | **Black NSD↑** | **White NSD↑** |
> |------|-----------------------|------------------|----------------|----------------|----------------|
> | **Cup**|                     |                   |                  |                |                |
> | |SAMed                       | 0.7222            | 0.6825           | 0.6871         | 0.7338         |
> | |SAMed+ADV                   | 0.7405            | 0.7045           | 0.6954         | 0.7537         |
> | |SAMed+GroupDRO              | 0.7415            | 0.7093           | 0.6942         | 0.7547         |
> | |**Ours (SAMed)**            | 0.7399            | 0.7038           | 0.7015         | 0.7518         |
> | |TransUNet                   | 0.9314            | 0.8845           | 0.8908         | 0.9449         |
> | |TransUNet+ADV               | 0.9208            | 0.8819           | 0.8817         | 0.9331         |
> | |TransUNet+GroupDRO          | 0.9285            | 0.8768           | 0.8873         | 0.9426         |
> | |**Ours (TransUNet)**        | 0.9262            | 0.8796           | 0.8850         | 0.9397         |
> | **Rim**  |                   |                   |                  |                |                |
> | |SAMed                       | 0.7483            | 0.7313           | 0.7215         | 0.7556         |
> | |SAMed+ADV                   | 0.8063            | 0.7694           | 0.7686         | 0.8182         |
> | |SAMed+GroupDRO              | 0.8078            | 0.7760           | 0.7696         | 0.8191         |
> | |**Ours (SAMed)**            | 0.8043            | 0.7732           | 0.7704         | 0.8146         |
> | |TransUNet                   | 0.9601            | 0.9326           | 0.9326         | 0.9688         |
> | |TransUNet+ADV               | 0.9554            | 0.9245           | 0.9221         | 0.9656         |
> | |TransUNet+GroupDRO          | 0.9596            | 0.9322           | 0.9307         | 0.9685         |
> | |**Ours (TransUNet)**        | 0.9581            | 0.9306           | 0.9315         | 0.9666         |
>
>
> **Add nnUNet.**
>
> We have included nnUNet in our segmentation benchmarks, which are detailed in the supplementary material. The table provided below presents results focusing on racial disparities. From the table, it is evident that nnUNet demonstrates marginally better performance compared to SAMed and TransUNet. However, it still exhibits significant performance disparities across different racial groups. This suggests that algorithmic unfairness is a pervasive issue in our proposed cup-disc segmentation, regardless of the choice of segmentation architectures.
>
>
> |       | Method | Overall ES-Dice↑ | Overall Dice↑ | Overall ES-IoU↑ | Overall IoU↑ | Asian Dice↑ | Black Dice↑ | White Dice↑ | Asian IoU↑ | Black IoU↑ | White IoU↑ |
> |-------|--------|------------------|---------------|-----------------|--------------|-------------|-------------|-------------|------------|------------|------------|
> | Cup   | nnUNet | 0.8625           | 0.8710        | 0.7704          | 0.7865       | 0.8675      | 0.8855      | 0.8697      | 0.7639     | 0.8035     | 0.7725     |
> | Rim   | nnUNet | 0.8003           | 0.8335        | 0.6959          | 0.7231       | 0.7930      | 0.7682      | 0.8490      | 0.6854     | 0.6639     | 0.7397     |
>
>
> **The dataset was released as npz format. Could you please also release the original format?**
>
> We will release the dataset with the original format in addition to the existing npz format of our database.
>
> **It would be great if you could release the trained models as well.**
>
> The trained checkpoints of our models have been released through our Github repository.
>
> **Where do you plan to host this benchmark? CodaLab could be a good platform.**
>
> We will co-host our benchmark/dataset using both Google Drive and CodaLab.
>
> **Reference:**
>
> [1] Kirillov, Alexander, et al. "Segment anything." arXiv preprint arXiv:2304.02643 (2023).
>
> [2] Zhang, Kaidong, and Dong Liu. "Customized segment anything model for medical image segmentation." arXiv preprint arXiv:2304.13785 (2023).
>
> [3] Chen, Jieneng, et al. "Transunet: Transformers make strong encoders for medical image segmentation." arXiv preprint arXiv:2102.04306 (2021).

---

### Official Review · Reviewer_5cUh · 2023-11-01

**Soundness:** 3 good
**Presentation:** 4 excellent
**Contribution:** 4 excellent
**Rating:** 6
**Confidence:** 3

**Summary:**

In this work, the authors introduced the new FairSeg dataset, designed to address fairness concerns in the domain of medical segmentation. Their innovative methodology centers on a fair error-bound scaling technique, which recalibrates the loss function by considering the upper error-bound within each identity group. Furthermore, they designed a new equity-scaled segmentation performance metric to facilitate fair comparisons between different fairness learning models for medical segmentation. Extensive experimentation underscores the efficacy of the fair error-bound scaling approach, demonstrating either superior or comparable fairness performance when compared to state-of-the-art fairness learning models. Furthermore, The related dataset and code are both made publicly accessible by the authors.

**Strengths:**

+ The paper is well-written and easy to follow.
+ The proposed framework is technically sound.
+ The experiments are comprehensive.

**Weaknesses:**

There is no visualization comparison between different methods.

**Questions:**

1. In equation (1), a parenthesis is missing in the formula.
2. The authors proposed a new the Dice loss with a novel Fair Error-Bound Scaling mechanism, however there are experiment results to show the differences between the new dice loss and common one.

---

> ### Author Response · Authors · 2023-11-22
> **Response to Reviewer 5cUh**
>
> Thank you very much for your review and the insightful comments.
>
>
> **There is no visualization comparison between different methods.**
>
> We have added a visualization comparison of segmentation results between our method and other competing approaches.
>
> **In equation (1), a parenthesis is missing in the formula.**
>
> We have addressed the formatting issue in the paper.
>
> **The authors proposed a new Dice loss with a novel Fair Error-Bound Scaling mechanism, however, there are experiment results to show the differences between the new dice loss and the common one.**
>
> As mentioned in Section 6.2 - “Training and Implementation Details”, the original SAMed and TransUNet are trained using cross entropy and the common dice losses. Hence, in Tables 1-5,  the results between SAMed vs SAMed (Ours) / TransUNet vs. TransUNet (Ours) are the performance comparisons for the new proposed dice loss and the common one.

---

### Official Review · Reviewer_Suah · 2023-11-06

**Soundness:** 3 good
**Presentation:** 3 good
**Contribution:** 4 excellent
**Rating:** 8
**Confidence:** 3

**Summary:**

This paper proposes a publicly available medical fairness segmentation dataset (FairSeg) that contains 10,000 subject samples of 2D SLO Fundus images. The paper also proposes equity-scaled segmentation performance metrics to facilitate fair comparisons.

**Strengths:**

1. The fairness concern is an important topic, especially in medical images and the lack of segmentation dataset is a big issue. The motivation of the proposed dataset is strong.

2. The dataset contains a large amount of segmentation ground truths (10,000) and is well evaluated by authors with several SOTA learning algorithms.

3. As described by the authors, the segmentation seems to undergo a rigorous process including a hand-graded annotation by a panel of five medical professionals after initial registration.

**Weaknesses:**

1. The accuracy of the Nifty reg needs to be investigated since it might not be the SOTA for image registration.

**Questions:**

1. Why validation set is not constructed/used in selecting models in training?

2. It would be helpful to report Hausdorff distance and average surface distance along with Dice to better evaluate the methods.

3. The details of how standard deviation is computed need to be elaborated. Is it computed across the mean of for each group?

4. How is the training/testing split performed? Is it just randomly sampled without considering sensitive attributes at patient level?

5. It would be helpful to discuss the importance of registration in preprocessing using NiftyReg.

---

> ### Author Response · Authors · 2023-11-22
> **Response to Reviewer Suah (Part 1)**
>
> Thank you very much for the supportive comments and valuable suggestions!
>
>
> **The accuracy of the Nifty reg needs to be investigated since it might not be the SOTA for image registration.**
>
> In Section 4, we mentioned that “It’s noteworthy that this registration operation demonstrates considerable precision in real-world scenarios, as evident from our empirical observations that highlight an approximate 80% success rate in registrations.”  During experiments, we have compared Niftyreg against the SOTA deep learning based approach [1] and a retinal image-based registration method [2]. We observed that NiftyReg is more robust than other registration methods. Upon further analysis, we have calculated that NiftyReg achieves an accuracy of roughly 82.4% in registration tasks. All the failed registration cases have been excluded by five professional clinician graders.
> Although there is a failure rate of about 20%, we are committed to ongoing exploration of state-of-the-art (SOTA) registration tools, including contemporary deep learning-based methods. Our aim is to enhance registration accuracy and, consequently, to release more images in our datasets during our future work.
>
> **Why validation set is not constructed/used in selecting models in training?**
>
> Given that our model is selected based on the last epoch of training. The finetuning of foundation models like SAM could be computationally expensive for both training and inference. This makes computing the validation accuracy every few epochs and selecting the checkpoints based on the accuracy is infeasible.
>
> To better facilitate future research, we have released an extra 500 images as the validation set in our codebase and dataset.
>
> **Report Hausdorff distance and average surface distance along with Dice.**
>
>
> Please see the table below for Hausdorff distance (HD95) and average surface distance (ASD) for race. We have included such metrics across all sensitive attributes in the supplementary material. From table below, we observed that the disparity between different demographic groups commonly existed regardless of its evaluation metrics, which further strengthened the significance of our proposed FairSeg dataset.
>
>
> |      | Method               | Overall HD95↓ | Asian HD95↓ | Black HD95↓ | White HD95↓ | Overall ASD↓ | Asian ASD↓ | Black ASD↓ | White ASD↓ |
> |------|----------------------|---------------|-------------|-------------|-------------|--------------|------------|------------|------------|
> | **Cup**  | SAMed                | 9.6231        | 11.0005     | 11.0142     | 9.1866      | 3.9650       | 4.6765     | 4.7743     | 3.7209     |
> |      | SAMed+ADV            | 9.4594        | 11.0170     | 11.1193     | 8.9479      | 4.0123       | 4.8623     | 4.9236     | 3.7321     |
> |      | SAMed+GroupDRO       | 9.4633        | 11.0891     | 11.0449     | 8.9603      | 3.9494       | 4.6462     | 4.8634     | 3.6855     |
> |      | Ours (SAMed)         | 9.4494        | 10.8376     | 11.1637     | 8.9453      | 3.9288       | 4.7042     | 4.8210     | 3.6606     |
> |      | TransUNet            | 4.7577        | 5.6983      | 5.5411      | 4.4937      | 2.0552       | 2.3692     | 2.4573     | 1.9382     |
> |      | TransUNet+ADV        | 5.0157        | 5.9379      | 5.8292      | 4.7476      | 2.1319       | 2.4693     | 2.4978     | 2.0198     |
> |      | TransUNet+GroupDRO   | 4.8195        | 5.6424      | 5.6753      | 4.5535      | 2.0615       | 2.3637     | 2.4952     | 1.9393     |
> |      | Ours (TransUNet)     | 4.9603        | 5.8818      | 5.7398      | 4.6992      | 2.1441       | 2.4641     | 2.5445     | 2.0268     |
> | **Rim**  | SAMed                | 9.9379        | 11.4859     | 11.5671     | 9.4337      | 3.9353       | 4.5013     | 4.5473     | 3.7476     |
> |      | SAMed+ADV            | 8.8316        | 10.5364     | 10.2364     | 8.3562      | 3.3523       | 3.9946     | 3.8975     | 3.1700     |
> |      | SAMed+GroupDRO       | 8.7609        | 10.4638     | 10.0592     | 8.3076      | 3.3473       | 3.8345     | 3.9499     | 3.1702     |
> |      | Ours (SAMed)         | 8.7930        | 10.3167     | 10.3407     | 8.3082      | 3.4174       | 4.0053     | 4.0608     | 3.2208     |
> |      | TransUNet            | 4.4404        | 5.4496      | 5.0893      | 4.1964      | 1.7534       | 1.9643     | 1.9941     | 1.6809     |
> |      | TransUNet+ADV        | 4.6301        | 5.6065      | 5.4390      | 4.3569      | 1.8110       | 2.1085     | 2.0885     | 1.7213     |
> |      | TransUNet+GroupDRO | 4.5268 | 5.4243 | 5.2290 | 4.2842 | 1.7498 | 1.9683 | 2.0017 | 1.6742 |
> |      | Ours (TransUNet)   | 4.5311 | 5.5251 | 5.2808 | 4.2682 | 1.8321 | 2.0642 | 2.0771 | 1.7564 |

---

> ### Author Response · Authors · 2023-11-22
> **Response to Reviewer Suah (Part 2)**
>
> **The details of how standard deviation is computed need to be elaborated. Is it computed across the mean of for each group?**
>
> The computational cost of multiple runs with large segmentation foundation models is high. Previous studies  [3, 4]  typically evaluate these models only once, without standard deviation, rather than conducting multiple runs of evaluations.
>
> **How is the training/testing split performed? Is it just randomly sampled without considering sensitive attributes at patient level?**
>
> The training and testing split are just randomly sampled, and our sample distribution of different sensitive attributes reflects the real-world clinical patient distribution.
>
> **It would be helpful to discuss the importance of registration in preprocessing using NiftyReg.**
>
> As mentioned above and in Section 4/Figure 1, to obtain a large-scale fundus dataset with high-quality pixel-wise annotation, we need to use the OCT machine to generate the OCT fundus images and their corresponding cup-disc mask. However, OCT machines are fairly expensive and less prevalent in primary care, therefore, we propose to migrate those annotations from 3D OCT to 2D SLO fundus for potentially broader impact in early-stage glaucoma screening in the primary care domains. In order to transfer the annotations from 3D OCT fundus images to 2D SLO fundus images, we need to register the image to align the two imaging modalities by comparing the characteristic features between the two fundus imaging modalities of the same patient.  The computed alignment matrix is then applied to the disc-cup masks of the OCT fundus images, aligning them to the SLO fundus images.
>
>
>
>
>
> **Reference:**
>
> [1] Hoopes, Andrew, et al. "Hypermorph: Amortized hyperparameter learning for image registration." Information Processing in Medical Imaging: 27th International Conference, IPMI 2021, Virtual Event, June 28–June 30, 2021, Proceedings 27. Springer International Publishing, 2021.
>
> [2] https://github.com/tobiaselze/oct_fundus_registration/tree/main
>
> [3] Kirillov, Alexander, et al. "Segment anything." arXiv preprint arXiv:2304.02643 (2023).
>
> [4] Zhang, Kaidong, and Dong Liu. "Customized segment anything model for medical image segmentation." arXiv preprint arXiv:2304.13785 (2023).

---

### Author Response · Authors · 2023-11-22
**General Response from Authors**

Dear AC and Reviewers,

We are truly grateful for the time and effort you have invested in reviewing our paper. Your constructive feedback has been invaluable in enhancing the quality of our work. We are particularly appreciative of your recognition of the importance of the new task (FairSeg) we have proposed and our comprehensive experiments.

In response to your comments and suggestions, we have made several significant updates to our manuscript, which we summarize as follows:

1. **Inclusion of More Segmentation Backbones**: In our revised version, we have incorporated nnUNet to further demonstrate that algorithmic unfairness is a prevalent issue in cup-disc segmentation tasks, regardless of the segmentation architecture employed.

2. **Expansion of Evaluation Metrics**: To provide a more thorough assessment of segmentation accuracy, we have added Normalized Surface Distance (NSD), Hausdorff Distance (HD95), and Average Surface Distance (ASD). Additionally, to better evaluate fairness, we have included the DPD and DEodds metrics.

3. **Enhanced Clarity in Writing**: We have made concerted efforts to clarify areas of the manuscript that reviewers found unclear or sought more information about. These amendments and additions are reflected in our revised submission.

We hope that these updates effectively address the concerns raised by the reviewers and clarify any ambiguities. We are eager to engage further and provide any additional information that may be required.

Sincerely,

The Authors

---

### Meta-Review · Area_Chair_YupL · 2023-12-11

**Metareview:**

This paper introduces the FairSeg dataset, a new benchmark to address fairness concerns in medical segmentation, garnering unanimous and consistent acceptance with scores of 6, 6, 8, and 8 across reviews. The common strengths highlighted in the four reviews center on the significant contribution of novel datasets tailored for medical image segmentation while integrating considerations for fairness. Particularly commendable is the technical soundness demonstrated through innovative methodologies, notably the introduction of a fair error-bound scaling technique that recalibrates the loss function by considering the upper error-bound within each identity group, complemented by the introduction of novel equity-scaled segmentation metrics facilitating fair model comparisons. The paper is appraised for its well-written presentation, technical rigor, and comprehensive experiments. Additionally, its commitment to open access for both dataset and code enhances reproducibility and fosters further research. Identified weaknesses include recurring criticisms related to the choice of evaluation metrics, with reviewers advocating for a more thorough exploration of alternatives and a clearer rationale behind baseline setting and evaluation choices. The absence of references to current works is also noted.

**Justification For Why Not Higher Score:**

As highlighted by the reviewers, although the paper addresses a critical medical benchmark with potential in the medical field, its broader scientific value, relevance, and significance to the diverse audience of ICLR may be somewhat constrained.

**Justification For Why Not Lower Score:**

This paper presents the FairSeg dataset, serving as a novel benchmark to tackle fairness concerns in medical segmentation. It achieves unanimous and consistent acceptance, earning scores of 6, 6, 8, and 8 across reviews. The merits are clearly evident— the paper is technically robust and garners praise from all reviewers.

---

### Decision · Program_Chairs · 2024-01-16

Accept (poster)